Molecular and morphological evidence for a new species of Leptopus (Phyllanthaceae) from Southeast Yunnan, China

Zhang Wenhua 1
Zhu Xinxin 2
Xue Bine 3
Liu Ende 4
Li Yuling 1
Yao Gang gyao@scau.edu.cn 1
1 College of Forestry and Landscape Architecture, South China Agricultural University , Guangzhou , Guangdong , China
2 College of Life Sciences, Xinyang Normal University , Xinyang , Henan , China
3 College of Horticulture and Landscape Architecture, Zhongkai University of Agriculture and Engineering , Guangzhou , Guangdong , China
4 Key Laboratory for Plant Diversity and Biogeography of East Asia, Kunming Institute of Botany , Kunming , Yunnan , China
Sosa Victoria
Electronic publication date: 2021 Aug 24
Publication date: 2021
Volume: 9
Electronic Location ID: e11989
Received 2021 May 17; Accepted 2021 Jul 26
Copyright: ©2021 Zhang et al.
Copyright year: 2021
Copyright holder: Zhang et al.
License: This is an open access article distributed under the terms of the Creative Commons Attribution License, which permits unrestricted use, distribution, reproduction and adaptation in any medium and for any purpose provided that it is properly attributed. For attribution, the original author(s), title, publication source (PeerJ) and either DOI or URL of the article must be cited.
License URL: https://creativecommons.org/licenses/by/4.0/

Keywords: Leptopus, Phyllanthaceae, Poranthereae, Taxonomy, Yunnan, China

Funding: The Natural Science Foundation of Guangdong Province, China 2019A1515011695 This work was supported by grant from the Natural Science Foundation of Guangdong Province, China (Grant no. 2019A1515011695 awarded to Gang Yao). The funders had no role in study design, data collection and analysis, decision to publish, or preparation of the manuscript.

==============================
Leptopus malipoensis, a new species of Phyllanthaceae from Southeast Yunnan Province, China, is described. The phylogenetic position of the new species within the genus Leptopus was analyzed based on nuclear ribosomal Internal Transcribed Spacer (nrITS) and plastid matK sequence data. The results show that L. malipoensis is highly supported to be the sister of L. fangdingianus (P. T. Li) Voronts. & Petra Hoff., a species endemic to western Guangxi Province, China. Morphologically, the new species can be distinguished from all known congeneric taxa by its long and slim branches usually pendulous or procumbent, some of its leaf laminas up to 15 cm long and 7 cm wide. It further differs from its sister species by its hirsute stems, leaves and pedicel of female flowers, longer pedicel of male flowers, 3-locular ovary and three styles. A key to all 10 accepted Leptopus species is provided.

Introduction

Leptopus Decne. is a small genus in the tribe Poranthereae, subfamily Phyllanthoideae of Phyllanthaceae and distributed mainly from the Caucasus to Malesia (Vorontsova & Hoffmann, 2009; Webster, 2014). In the latest taxonomic revision of Leptopus, nine species were accepted and six among them were recorded in China (Vorontsova & Hoffmann, 2009). Results from molecular phylogenetic analyses showed that Leptopus was sister to Actephila Blume with high support (Vorontsova et al., 2007; Vorontsova & Hoffmann, 2008). Morphologically, the genus Leptopus can be distinguished from Actephila by its disc evidently lobed (vs. annular), fruit less than 10 mm in diameter (vs. more than 10 mm in diameter), exocarp adnate to endocarp (vs. free from endocarp) and mature seed with copious endosperm (vs. seed without endosperm) (Li et al., 2008). Results from recent biochemical studies suggested that chemical components extracted from Leptopus species have potential to be used in treatment of cancer, diabetes mellitus and hyperlipemia (Qi et al., 2021; Rahman et al., 2021).

During the field investigations in Malipo Hsien, southeast Yunnan Province of China, in March 2018, two of the authors (E.D. Liu and X.X. Zhu) collected a Phyllanthaceae specimen belonging to Leptopus, and the same species was recollected in the same locality by another author (G. Yao) in July 2020. The species is very different from all the other members of Leptopus in habit and morphology. After detailed morphological investigation and molecular phylogenetic analyses of Leptopus, it was concluded that the specimens represent a species that is new to science and formally described here.

Materials & Methods

Ethics statements

The collection location of the new species reported in this study is outside any natural conservation area and no specific permissions were required for the location. Since this species is currently undescribed, it is not currently included in the China Species Red List (Wang & Xie, 2004). However, due to conservation concerns and lack of habitat protection, exact locality coordinates have been withheld from our published specimen records. Our field studies did not involve any endangered or protected species. No specific permits were required for the present study.

Nomenclature

The electronic version of this article in Portable Document Format (PDF) will represent a published work according to the International Code of Nomenclature for algae, fungi, and plants (ICN), and hence the new names contained in the electronic version are effectively published under that Code from the electronic edition alone. In addition, new names contained in this work which have been issued with identifiers by IPNI will eventually be made available to the Global Names Index. The IPNI can be accessed and the associated information contained in this publication viewed through any standard web browser by using the web address “http://ipni.org/”. The online version of this work is archived and available from the following digital repositories: PeerJ, PubMed Central, and CLOCKSS.

Material collection

Flowering and fruiting specimens of the new species were collected in the mountain area, in Nandong to Bajiaoping, Laoshan, Malipo Hsien of Yunnan Province, China, for morphological study. Leaf materials for DNA extraction were collected and dried using silica gel in the field.

Morphological study

Specimens of Leptopus deposited in the herbaria GXMG, IBK, IBSC, KUN and PE were carefully examined for the present study. Field investigations of Chinese Leptopus species were also conducted in recent years. Morphological characters of stems, leaves, flowers and fruits of relevant species were photographed and measured. In addition, morphological comparisons between the new species and all the nine Leptopus species accepted by Vorontsova & Hoffmann (2009) were also conducted.

Phylogenetic study

Phylogenetic relationships among species of Leptopus were previously investigated by Vorontsova et al. (2007) based on analysis of two DNA markers, viz. the nuclear ribosomal Internal Transcribed Spacer (nrITS) and plastid matK. To resolve the relationships of the new species, a phylogenetic study of the genus Leptopus was performed, based on analyses of the two above mentioned DNA markers. Other DNA sequences of Leptopus species included in Vorontsova et al. (2007) were obtained from GenBank (https://www.ncbi.nlm.nih.gov/) and used in the present phylogenetic analyses. Three species of Leptopus (viz. L. colchicus (Fisch. & C.A. Mey. ex Boiss.) Pojark., L. calcareus (Ridl.) Pojark. and L. esquirolii (H. Lév.) P.T. Li) sampled in Vorontsova et al.’s (2007) phylogenetic study had been reduced to the synonyms of other species in their subsequent taxonomic revision of Leptopus (Vorontsova & Hoffmann, 2009). Thus, DNA sequences of the above mentioned three species were not included in the present phylogenetic analyses. Additionally, outgroups were selected from the other seven genera of the tribe Poranthereae (Actephila Blume, Andrachne L., Meineckia Baill., Notoleptopus Voronts. & Petra Hoffm., Phyllanthopsis (Scheele) Voronts. & Petra Hoffm., Pseudophyllanthus (Müll. Arg.) Voronts. & Petra Hoffm. and Poranthera Rudge) and the genus Heywoodia Sim of the tribe Wielandieae, based on previously published phylogenetic frameworks (Kathriarachchi et al., 2005; Vorontsova et al., 2007). DNA sequences of outgroups were also downloaded from GenBank. Detailed information about the species sampled and DNA sequences are provided in Table 1.

Table 1 Sequences information for all samples used in the present study.

Sequences newly generated in this study are marked in bold.

Taxon	nrITS	matK	
Leptopus australis (Zoll. & Moritzi) Pojark.	AM745811	AM745812	
Leptopus australis (Zoll. & Moritzi) Pojark.	AM745813	AM745814	
Leptopus chinensis (Bunge) Pojark.	MN722097	MN722149	
Leptopus chinensis (Bunge) Pojark.	MH710764	MH659095	
Leptopus chinensis (Bunge) Pojark.	AM745819	AM745820	
Leptopus chinensis (Bunge) Pojark.	AM745821	AM745822	
Leptopus clarkei (Hook. f.) Pojark.	AM745938	AM745939	
Leptopus clarkei (Hook. f.) Pojark.	AM745940	AM745941	
Leptopus cordifolius Decne.	AM745826	AY552433	
Leptopus cordifolius Decne.	AM745827	AM745828	
Leptopus cordifolius Decne.	AM745829	AY552433	
Leptopus fangdingianus (P.T. Li) Voronts. & Petra Hoffm.	AM745809	AM745810	
Leptopus malipoensis W.H. Zhang & Gang Yao	MW962203	MZ062211	
Actephila albidula Gagnep.	AM745910	AM745911	
Actephila collinsiae W. Hunter ex Craib	AM745912	AM745913	
Actephila sessilifolia Benth.	AM745931	AM745932	
Andrachne ephemera M.G. Gilbert	AM745767	AM745768	
Andrachne fruticulosa Boiss.	AM745773	AM745774	
Andrachne microphylla (Lam.) Baill.	AM745787	AM745788	
Andrachne telephioides L.	AM745802	AM745803	
Heywoodia lucens Sim	AM745935	AM745937	
Meineckia acuminata (Verdc.) J.F. Brunel	AM745894	AM745895	
Meineckia humbertii G.L. Webster	AM745846	AM745847	
Notoleptopus decaisnei (Benth.) Voronts. & Petra Hoffm.	AM745830	AM745831	
Notoleptopus decaisnei (Benth.) Voronts. & Petra Hoffm.	AM745832	AM745833	
Phyllanthopsis arida (Warnock & M.C. Johnst.) Voronts. & Petra Hoffm.	AM745762	AM745763	
Phyllanthopsis phyllanthoides (Nutt.) Voronts. & Petra Hoffm.	AM745836	AM745837	
Poranthera corymbosa Brongn.	AM745872	AM745873	
Poranthera triandra J.M. Black	AM745892	AM745893	
Pseudophyllanthus ovalis (E. Mey. ex Sond.) Voronts. & Petra Hoffm.	AM745789	AY830260	
Pseudophyllanthus ovalis (E. Mey. ex Sond.) Voronts. & Petra Hoffm.	AM745790	AM745791	

Total DNA of the new species was extracted from silica gel-dried leaves (voucher specimen: G. Yao YGYN2020071501, IBSC). DNA was sheared into ca. 500 bp fragments using a TurePrepTM DNA Library Prep Kit V2 for Illumina, following the manufacturer’s manual (Vazyme Biotech Co., Ltd., Nanjing, China) and then sequenced from both ends of 150 bp fragments on the Illumina HiSeq 2500 platform (Illumina, San Diego, CA, USA) at BGI Genomics (BGO-Shenzhen, China). About 3 Gb of raw data was generated. Plastid and the nrITS sequence reads were assembled using the software GetOrganelle (Jin et al., 2020), with the reference plastid genome of Glochidion chodoense C.S. Lee & Im (GenBank accession number: NC_042906) and nrITS sequence of Leptopus chinensis (Bunge) Pojark. (GenBank accession number: MH710764), respectively. Genes in the plastid genome obtained were annotated in the software PGA (Qu et al., 2019). The matK sequence was then extracted from the assembled whole plastid genome.

Sequences were aligned using MAFFT v. 7.221 (Katoh & Standley, 2013) and then three data sets were constructed: the matK dataset, the nrITS dataset and the combined dataset (including matK and nrITS). All the three datasets were analyzed using two approaches: Bayesian Inference (BI) and Maximum Likelihood (ML) were conducted using MrBayes v. 3.2.6 (Ronquist & Huelsenbeck, 2003) and RAxML (Stamatakis, 2006), respectively. The models of nucleotide substitution of the two DNA markers used were selected under the Akaike Information Criterion (AIC) using jModeTest v. 3.7 (Posada, 2008): TVM+G for matK and GTR+I+G for nrITS. Detailed information about the parameter setting in BI and ML analyses referred the phylogenetic analyses conducted in Yao et al. (in press), except that each of Markov Chain Monte Carlo (MCMC) analysis was run for 10,000,000 generations, sampling every 500 generations. Number of generations for the datasets were sufficient, because the average standard deviations (SD) of split frequencies for the datasets were all below 0.01, and effective sample sizes (ESS) of all parameters were over 200 as evaluated in Tracer v. 1.6 (Rambaut, Suchard & Drummond, 2014). The first 25% of the trees obtained in BI analyses were discarded as burn-in and then posterior probabilities (PP) were determined from the posterior distribution.

Results

Phylogenetic analysis

The matK dataset, nrITS dataset and combined dataset alignments contained 2,000 bp, 812 bp and 2,812 bp, respectively. Conflicted topologies were found between the matK and nrITS frameworks (Fig. 1), but relevant conflicted phylogenetic nodes were all poorly supported in analyses of the nrITS dataset (Fig. 1B). Phylogenetic relationships derived from the combined dataset were much better resolved compared with those obtained from analyses based on the other two datasets, and phylogenetic relationships among Leptopus species sampled here were all resolved with high support values (Fig. 2). Thus we focus on describing phylogenetic relationships based on the result derived from the combined dataset.

Figure 1 Maximum likelihood (ML) trees of Leptopus based on analysis of matK (A) and nrITS (B).

Maximum likelihood (ML) trees of Leptopus and its relatives inferred from the matK dataset (A) and nrITS dataset (B). Bootstrap (BS) value ≥ 50% in ML analysis and posterior probability (PP) ≥ 0.50 in Bayesian inference (BI) is indicated on the left and right of slanting bar associated with phylogenetic node, respectively. Dashes denote that the phylogenetic node associated was not supported or the BS value is < 50% in ML analysis or PP> 0.50 in BI. The crown node of Leptopus is shown by the arrowhead.

Figure 2 Maximum likelihood (ML) tree of Leptopus and its relatives inferred from the combined data set (including nrITS and matK).

Bootstrap (BS) value in ML analysis and posterior probability (PP) in Bayesian inference (BI) is indicated on the left and right of slanting bar associated with phylogenetic node, respectively. The crown node of Leptopus is shown by the arrowhead.

Phylogenetic results showed that the genus Leptopus was sister to Actephila with high support values (Bootstrap (BS) = 100%, PP = 1.00), and the monophyly of Leptopus was strongly supported (BS = 100%, PP = 1.00). Within Leptopus, the species L. australis (Zoll. & Moritzi) Pojark. represents the earliest divergent taxa in this genus. The new species is strongly supported as the sister of L. fangdingianus (P.T. Li) Voronts. & Petra Hoffm. (BS = 100%, PP = 1.00) and this pair in turn sister to L. clarkei (Hook. f.) Pojark. (BS = 100%, PP = 1.00). Leptopus chinensis (Bunge) Pojark. and L. cordifolius Decne. formed a sister clade (BS = 100%, PP = 1.00) and closely related to the (new species-L. fangdingianus)-L. clarkei clade with strong support (BS = 100%, PP = 1.00). Furthermore, the sister relationship between the new species and L. fangdingianus was highly supported in both of the matK (Fig. 1A) and nrITS (Fig. 1B) analyses.

Figure 3 Leptopus malipoensis (from the type locality).

(A–B & E–F) habit; (C) adaxial side of leaves; (D) abaxial side of leaves. (Photo credit: Gang Yao and Xin-Xin Zhu).

Figure 4 Leptopus malipoensis (from the type locality).

(A) Female flower; (B) male flower; (C) young fruit; (D) young leaf, stem and female flower with pedicel; (E) stem and male flower with pedicel; (F) stem and young fruits with fruiting pedicels; (G) fruit and fruiting pedicel; (H) fruit; (I) fruit transection; (J) capsule with fungus on its surface; (K) seeds. (Photo credit: Gang Yao and Xin-Xin Zhu).

Morphological comparisons

A detailed morphological comparison between the new species and other members of the genus Leptopus was conducted, and field images of the new species are provided in Figs. 3–4. Morphologically, the new species has lobed discs in both the pistillate (Fig. 4A) and staminate flowers (Fig. 4B), fruits 4–6 mm in diameter (Figs. 4I & 4J) and mature seeds with copious endosperm (Fig. 4I). These characters are congruent with its placement in the genus Leptopus. However, the species has pendulous or procumbent stems (Figs. 3A–3B) and some of its leaf laminas could up to 15 cm long and 7 cm wide. These characters make it distinct from all the other Leptopus members, which usually have ascendant or erect stems and smaller leaves less than 10 cm long and 5 cm wide. The procumbent habit is also recorded in L. clarkei, a species widely distributed from southern China, extending west to Assam and Burma, and south to northern Vietnam (Vorontsova & Hoffmann, 2009). The species is also closely related to the collective clade formed by the new species and L. fangdingianus in the phylogenetic analyses (Figs. 1 and 2). However, the new species differs from L. clarkei by having straight and not ribbed stems (Figs. 3C–3F) (vs. flexuous and longitudinally strongly ribbed stems), laminas up to 15 cm long and 7 cm wide (vs. leaf laminas all less than 10 cm long and 3 cm wide), leaves hirsute on both surfaces (Figs. 3C–3E & 4D) (vs. glabrous adaxially and glabrous to sparsely hirsute abaxially), margin of leaves densely hirsute (Figs. 3C–3E & 4D) (vs. glabrous to hirsute). While the new species differs from its sister L. fangdingianus by its leaves hirsute on both surfaces (Figs. 3C–3E & 4D) (vs. glabrous to sparsely hirsute on both surfaces), pedicel of male flowers 10–25 mm long (vs. usually less than 10 mm long), pedicel of female flowers hirsute (Fig. 4D) (vs. glabrous), ovary 3-locular (vs. 4–5) and styles 3 (vs. 4–5).

Morphologically, the new species also can be easily distinguished from the remaining four congeneric species which were not included in the present phylogenetic analyses [viz., L. emicans (Dunn) Pojark., L. hainanensis (Merr. & Chun) Pojark., L. pachyphyllus X.X. Chen and L. robinsonii Airy Shaw], based on its pendulous or procumbent stems and branches. Additionally, the new species has hirsute indumentum, unribbed branches, chartaceous leaves with 4 (rarely 3 or 5) pairs of secondary veins. In contrast, L. emicans has glabrous longitudinally ribbed branches and 8–10 pairs of secondary veins in leaf laminas. Leptopus hainanensis also has glabrous branches and leaves, as well as 2–3 pairs of secondary veins in leaves. The other two species L. pachyphyllus and L. robinsonii both have glabrous and bilaterally flattened branches, and the leaves of L. pachyphyllus are glabrous on both sides and almost succulent.

Taxonomic treatment

Leptopus malipoensis W.H. Zhang & Gang Yao, sp. nov. (Figs. 3–4)	

IPNI

Type. CHINA. Yunnan Province, Wenshan State, Malipo Hsien, Laoshan, Nandong to Bajiaoping, on rocky slopes near the roadsides of the semi-shady forests, at the elevation of ca. 1,200 m, 15 July 2020, G. Yao YGYN2020071501 (holotype: IBSC; isotypes: KUN, CANT).

Diagnosis. The species is similar to L. fangdingianus (P.T. Li) Voronts. & Petra Hoffm. in general morphology, but differs from the latter by its procumbent habit with long and slim branches usually pendulous, some of its leaves could up to 15 cm long and 7 cm wide, hirsute stems, leaves and pedicel of female flowers, longer pedicel of male flowers, 3-locular ovary and 3 styles.

Description. Shrub, monoecious. Stems straight, terete, hirsute; branchlets slender, sometimes up to 1.5 m long, usually pendulous or procumbent, hirsute, sometimes rooting on branchlets. Leaves alternate, chartaceous, elliptic to ovate, (2)5–13(15) cm long and (1.2)3–5.5(7) cm wide, 1.5–2.3 times longer than wide, both surfaces hirsute, densely hirsute when young, margin densely hirsute, base cuneate to round, apex acuminate; midvein adaxially impressed, abaxially raised; secondary veins usually 4 pairs, rarely 3 or 5 pairs, adaxially slightly impressed, abaxially raised, obliquely ascending, sometimes arcuately anastomosing near margins. Petiole 4–12 mm long, hirsute. Stipules triangular, very small, less than 0.5 mm in both of length and width. Inflorescences unisexual or bisexual, axillary, fasciculate. Bracts narrowly triangular, 1–2 mm long and 0.2–0.5 mm wide. Staminate flowers 1–3 per fascicle, ca. 3 mm in diameter, light yellow to slightly green; pedicel 10–25 mm long, glabrous; sepals 5, ca. 2–2.5 mm long and ca. 1–1.3 mm wide, oblong, apically round, adaxially glabrous, abaxially sparsely hirsute, 0–3-veined; petals 5, shorter than sepals, clavate to slightly linear, alternating with sepals; disc extrastaminal with 5 contiguous regular segments deeply bilobed for 1/3–1/2 of length, apices of lobes truncate to rounded; stamens 5, opposite sepals; filaments 5, free; anthers 5, longitudinally dehiscent. Pistillate flowers usually 1 per fascicle, 3.5–4 mm in diameter; pedicels usually 15–28 mm long, sparsely hirsute, apically dilated; sepals 5, 2–3 mm long and ca. 1–1.5 mm wide, oblong to ovate-triangular, apically acute to round, adaxially glabrous, abaxially glabrous or sparsely hirsute, usually 0–5-veined; petals 5, shorter than sepals, linear, alternating with sepals; disc with 5 contiguous regular segments deeply biobed for ca. 1/2 of length, apices of lobes truncate to rounded; ovary 3-locular, globose, glabrous, ovules 2 per locule; styles 3, free, apex bifid to base, lobes usually recurved, stigmas dilated to capitate. Fruiting pedicel 2–3.3 cm long, hirsute; Fruit a capsule, dehiscent into 3 2-valved cocci when mature, depressed globose, smooth, glabrous or sparsely hirsute when young, 4–6 mm in diameter, 2.5–3 mm high, persistent sepals oblong; seeds 6, 2 per locule, brown to dark-brown, hemispheric or laterally compressed, ca. 2.5 mm long and 2 mm wide, endosperm fleshy, lacking appendages.

Etymology.Leptopus malipoensis is named after its type locality, Malipo Hsien. Malipo Hsien is a hotspot for biodiversity research in Yunnan Province, China, and many new species have been described recently from there, e.g., Bredia malipoensis D. H. Peng, S. Jin Zeng & Z.Y. Wen in Melastomataceae (Wen et al., 2019), Habenaria malipoensis Q. Liu & W.L. Zhang (Zhang et al., 2017) and Vanda malipoensis L.H. Zou, J.X. Huang & Z.J. Liu (Zou, Liu & Huang, 2014) in Orchidaceae, Primulina malipoensis L.H. Yang & M. Kang in Gesneriaceae (Yang et al., 2018), and Salacia malipoensis X.D. Ma & J.Y. Shen in Celastraceae (Ma et al., 2020).

Phenology: Flowering in April to August, and fruiting in May to October.

Paratype: CHINA. Yunnan Province, Wenshan State, Malipo Hsien, Laoshan, Nandong to Bajiaoping, under the semi-shady forests, at the elevation of 1171 m, 5 March 2018, Z.D. Wei, F.Z. Shangguan, X.X. Zhu et al. LiuED8755 (KUN).

Distribution and habitat: The species is known only from its type locality, Malipo Hsien in southeast Yunnan Province, China (Fig. 5).

Figure 5 Distribution of Leptopus malipoensis (blue square) and L. fangdingianus (red circular).

Habitat. The species grows on rocky slopes near the roadsides of the semi-shady forests or under the semi-shady forests, in limestone environments, at an elevation of 1170 −1200 m.

Chinese name. Ma Li Po Que She Mu (麻栗坡雀舌木).

Discussion

As recorded in previous taxonomic literature, nine Leptopus species were accepted and six of them were recorded in China (Li et al., 2008; Vorontsova & Hoffmann, 2009). In the present study, morphological and molecular evidence supported the recognition of the tenth species of Leptopus. The close relationship between the new species and L. fangdingianus is expected because they share some biogeographic and ecological similarities. The distribution areas of two species are adjacent to each other (Fig. 5). The new species is endemic to southeast Yunnan Province, China, while L. fangdingianus is endemic to western Guangxi Province, China. Both areas are characterized by a karst environment.

After the taxonomic revision of the genus Leptopus (Vorontsova & Hoffmann, 2009), Adhikari, Chaudhary & Ghimire (2010) described the new species Leptopus nepalensis B. Adhikari, R.P. Chaudhary & S.K. Ghimire from Nepal, based on the specimen B. Adhikari 224 deposited in the herbarium TUCH. On the basis of the morphological description provided by Adhikari, Chaudhary & Ghimire (2010), L. nepalensis is characterized by its 6 petals in two whorls, 6 stamens (or 3 as observed from the illustration drawn based on the holotype), and 3 styles connate into a column up to about halfway. However, all of these characters are very different from those of the genus Leptopus as currently circumscribed (Vorontsova & Hoffmann, 2009; Webster, 2014). Thus the species L. nepalensis is not accepted in the genus Leptopus in the present study.

Key to species of Leptopus, modified from Vorontsova & Hoffmann (2009)

1a. Leaf laminas coriaceous, almost succulent; endemic to Guangxi Province, China …………………………………………………………………L. pachyphyllus	
1b. Leaf laminas membranaceous to thick chartaceous, never succulent …..……. 2	
2a. Leaf laminas with 8–10 visible pairs of secondary veins; fruit strongly reticulate; seed with orange micropylar appendage ………………………………….. L. emicans	
2b. Leaf laminas with 0–6 (7) visible pairs of secondary veins; fruit smooth to faintly reticulate; seed without appendage …….…………………..…………………3	
3a. Ascendant herb to subshrub up to 0.5 m high; pedicels of female flowers 2–5 mm in flower, 5–9 mm in mature fruit ……………………………………L. australis	
3b. Erect to procumbent herb or shrub 0.5–4 m; pedicels of female flowers 5–30 mm in flower, 7–36 mm in mature fruit ……………………………………………. 4	
4a. Pedicels of male flowers less than 3 mm in length ……………………………5	
4b. Pedicels of male flowers more than 3 mm in length, and usually up to 10 mm or longer ……………………………………………………………………………….. 6	
5a. Branches white to light brown; male petals, filaments and styles mostly glabrous; seeds smooth; endemic to Hainan Province, China ………. L. hainanensis	
5b. Branches reddish; male petals, filaments and styles hirsute; seeds transversely to irregularly ridged, sometimes pitted; endemic to Khanh Hoa Province, Vietnam …………………………………………………………….……L. robinsonii	
6a. Stem flexuous; branches strongly ribbed ……..…….…………….….. L. clarkei	
6b. Stem straight; branches terete to moderately ribbed ………………………….. 7	
7a. Petioles of mature leaves at least 1/4 of leaf lamina length; leaf apex rounded ………………………………………….……….……………... L. cordifolius	
7b. Petioles of mature leaves 1/10 −1/6 of leaf lamina length; leaf apex not rounded (except several specimens of L. chinensis) ………………….…..…………. 8	
8a. Twigs longitudinally ribbed, or sometimes terete; leaf laminas usually 0.8–3 cm long (never longer than 5 cm) and 0.4–1.5 wide ……………….……….. L. chinensis	
8b. Twigs never longitudinally ribbed; leaf laminas usually longer than 5 cm and wider than 2.5 cm ……………...…………………………………………………….. 3	
9a. Stems and branchlets usually pendulous or procumbent; ovary 3-locular; styles 3; endemic to Yunnan Province, China ……………………………….. L. malipoensis	
9b. Stems and branchlets ascendant or erect; ovary 4–5-locular; styles 4–5; endemic to Guangxi Province, China …………………….………………….. L. fangdingianus	

Supplemental Information

Supplemental Information 1 ITS sequence of the new species described

Click here for additional data file.

Supplemental Information 2 Matrix of matK

Click here for additional data file.

Supplemental Information 3 Matrix of matK & nrITS

Click here for additional data file.

Supplemental Information 4 Matrix of nrITS

Click here for additional data file.

Supplemental Information 5 matK sequence of the new species described

Click here for additional data file.

Supplemental Information 6 Raw data for plant measurements

Click here for additional data file.

The authors are grateful to the editor and three reviewers for helpful comments on our manuscript, to the curators and staff of the herbaria GXMG, IBK, IBSC, KUN and PE for hosting our visits or providing images of specimens, and to Mr. Long in Malipo Hsien, Laoshan, Yaowang Valley for help in field investigations.

Additional Information and Declarations

Competing Interests

Author Contributions

Field Study Permissions

Data Availability

New Species Registration

The authors declare there are no competing interests.

Wenhua Zhang performed the experiments, prepared figures and/or tables, authored or reviewed drafts of the paper, and approved the final draft.

Xinxin Zhu conceived and designed the experiments, authored or reviewed drafts of the paper, and approved the final draft.

Bine Xue performed the experiments, analyzed the data, prepared figures and/or tables, authored or reviewed drafts of the paper, and approved the final draft.

Ende Liu performed the experiments, authored or reviewed drafts of the paper, and approved the final draft.

Yuling Li performed the experiments, analyzed the data, authored or reviewed drafts of the paper, and approved the final draft.

Gang Yao conceived and designed the experiments, prepared figures and/or tables, authored or reviewed drafts of the paper, and approved the final draft.

The following information was supplied relating to field study approvals (i.e., approving body and any reference numbers):

The collection location of the new species reported in this study is outside any natural conservation area and no specific permissions were required for the location.

The following information was supplied regarding data availability:

The DNA sequences analyzed in our study are available in Table 1 and at NCBI: MW962203 and MZ062211.

Information about the voucher specimen (G. Yao YGYN2020071501, IBSC) used in molecular phylogenetic analyses is available in our article.

The following information was supplied regarding the registration of a newly described species:

Leptopus malipoensis W.H. Zhang & Gang Yao, sp. nov. LSID: 77218863-1.

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
