# Peer review of "Molecular and morphological evidence for a new species of Leptopus (Phyllanthaceae) from Southeast Yunnan, China"

_PeerJ, doi:10.7717/peerj.11989_

## Round 0.1 · original submission · Major Revisions

The three reviewers coincided that the paper needs a good revision on the English and in addition, some sections need to clarify issues indicated below, among them, methods need more detail. Also, it is stressed that the discussion section is lacking. One of the reviewers kindly reviewed these issues and they are included in the attached file.

·

Basic reporting

The English needs some improvement, I indicated a few instances. Overall quite good English (but I am not a native speaker).

Literature references are fine.

Overall the structure is fine, but I miss a section Discussion. The conflict in phylogenetic results between both markers can be discussed and the combined results can be compared with those of Vorontsova et al.'s phylogeny (which are likely the same, as most are their data); here you can also explain the difference in number of species between their and your data. In the discussion, the Indian species, likely not a Leptopus, can be discussed, now at the end of the manuscript.

Experimental design

It is difficult to comment on the molecular work as all used techniques are only represented by references, idem with the phylogenetic analysis. I expected at least a discussion for the Bayesian analysis about sufficient cycles and high enough scores with ESS values (program Tracer). Therefore, I think that adding the programs used and settings of the phylogenetic analyses will be an improvement, not just a simple reference.

For the rest fine..

Validity of the findings

The findings seem fine.

Additional comments

It is a short and good paper, showing that the new species is morphologically and phylogenetically distinct. The key also demonstrates even better than a discussion that the species is different.

Only you put a lot of focus on the different habit. But bear in mind that that is field character, generally not visible on herbarium specimens nor mentioned on the labels. Thus you cannot be certain that all other species have a different habit!

·

Basic reporting

The English while mostly clear and unambiguous has many grammar problems, and a few typos. I have flagged and/or suggested edits in this regard as tracked changes in an attached version of the manuscript. The literature cited covers all relevant materials related to the current taxonomy of Leptopus. A mention of the medicinal/biochemical studies in Leptopus could be added to make the reader aware of this aspect to the genus, even if not strictly relevant to this systematics study.

The full alignments as used for the phylogenetic analyses need to be made available, either as supplements here or in public databases such as Dryad. The analyses can’t be replicated without the alignments and complex alignments (likely the case for ITS) are difficult to replicate without knowing minute starting details. The fasta files made available at review only include the sequences of the new taxon and no context of the alignment. The disposition of the raw data for the sequence assemblies is unclear and preferably it would be deposited in an archive such as the SRA. This is especially relevant if long multi-locus assemblies are presented (see below) or if the remainder of the genome skim data is being ‘thrown away’ when it could be mined for other purposes by future investigators. The authors should be consistent on how they phrase the DNA they use for phylogenetic analyses; they use the term “fragment” or “DNA region”, but genes, genetic markers, or loci would be more appropriate. The figures are relevant, clear, and ample, and the 2 plates of live-plant photographs are equivalent to a botanical illustration in showing all the diagnostic parts. The support values on the phylogeny graphics should be carefully positioned so as not to touch or overlap the branch lines; for those on short nodes the numbers could be moved to underneath node. The multiple accessions for some species need to have identifiers (e.g., Leptopus australis 1, Leptopus australis 2) to tie positions in trees to accessions in Table 1. The legend for Fig.4K should specify units for scale (presumably each tic is a mm). I do not understand why Fig.4J is moldy – is this fruit taken in the wild and typical, or is it a poorly preserved specimen. Dot leaders in keys should be uniform to allow species to line up. Legend for Fig.5: “red circular” should be ‘red circle’.

Experimental design

The manuscript presents original primary research on a newly discovered species that is within the scope of the journal. The Methods associated with the DNA sequence generation and phylogenetic analyses are incompletely presented. In particular, how the data was generated is not detailed in this text (i.e., how were libraries prepared, what sequencing platform, read length, how many reads, how were the reads filtered, what was the coverage of the assemblies). It is not sufficient to simply cite this all under “Zeng et al. (2018)” which likely has many details different and slight technology changes from what the authors used here. It is also not sufficient to cite away most details of phylogenetic analyses under Yao et al. (2020); minimally the run parameters need to be specified here and, burn-in for BI. Were the data sets partitioned in any way? The Methods discuss using genome skimming to yield a full cp genome, and it is a lost opportunity not to present that genome with this manuscript, rather than just extract a single gene. Similarly, the reads should allow the assembly of the full rDNA repeat (ETS-26s) while only ITS was extracted here. Presenting those full sequences with some minimal descriptive aspects (length of cp genome, number of genes, etc.) would be a valuable addition to this manuscript and it remains odd to describe assembling and annotating a plastid genome but show nothing of it. If a full cp genome is to be presented elsewhere (such as a data release note in Mitochondrial DNA: Resources), then those intentions should be so indicated. “nrITS” can just be referred to as ITS.

Validity of the findings

The new species is validly presented (i.e., type designated, diagnosis) according to nomenclature rules and the plant itself is distinctive (both morphologically and in sequence divergence) enough within Leptopus to be considered a new species by a specialist taxonomist such as this reviewer. The phylogenetic evidence is sound; those analyses appear sufficient, but as I noted before the details on their quality rely on citation rather than explicit statements. The authors make some comparisons (morphology and close phylogenetic relationships) between the new species and L. fangdingianus, but they should do more to address biogeographic and ecological similarities. It’s striking that both these species are very rare, live relatively near each other, and live in similar habitats. Generally, the biogeography of Leptopus with its wide range across Eurasia (in both temperate and tropical regions), and mix of narrow endemics and broadly distributed species is interesting. It is unfortunate that not all species have been sampled for a phylogeny to enable greater evolutionary and biogeographic inferences.

As a plant with a narrow distribution and given that the authors have visited the type locality, it would be valuable to provide a conservation assessment (following IUCN criteria and categories). Was the new species brought into cultivation? The specimen localities are vague and could use coordinates; if conservation status is a concern in disclosing locality details, then the accuracy of those coordinates could be limited for publication. The reader should be able take those coordinates and use GoogleEarth to look at imagery of the type locality or plot its distribution without having to extrapolate from the described locality. The plant description could use more details and in particular is incomplete on measurements of described structures (i.e., shrub height, length and width of stipules, sepals, petals, disc-segments, stamens/filaments/anthers, styles); it should more closely match those details in Vorontsova & Hoffmann (2009). What is leaf venation type? Explicitly state 2 ovules and/or seeds per locule. What floral parts are persistent in fruit (i.e., sepals, styles)? Fruit type should be specified (presumably an explosive schizocarp splitting into 3 equal 2-valved mericarps). Describe columella and fruit valves after dehiscence. Having technical details that are parallel to the recent monograph on Leptopus is important for this new taxon as few specimens are available in 3 Chinese institutions.

Additional comments

This is a distinctive new species, for which the authors provide all the ingredients for a thoughtful taxonomic paper (description, revised key to species, phylogeny, illustrations, map). The “Custom Checks” for DNA, field study and new species have been followed as much as the state of the manuscript allows; the 2 new sequences are not publicly released yet (but have GenBank numbers and those data are also as supplements), and the LSID number must await ms. acceptance. It remains surprising that the authors don’t make broader use of their ample molecular data. Even though they need only a small amount of orthologous data to insert their new species in prior phylogenetic frameworks to determine relationships, describing more of their genomic sequencing efforts would make the manuscript more modern in approach and allow the authors’ results to be incorporated into broader future studies.

Reviewer 3 ·

Basic reporting

no comment

Experimental design

no comment

Validity of the findings

no comment

Additional comments

The manuscript has interesting results and appropriately conducted. However, there are some information that need to be reconsidered. The major comments are as following and please see some minor comment and suggestion in the annotated MS.
1. In the Introduction.
Introduction should include all important information for the studied genera, thus information of Leptopus nepalensis should be mentioned in the Introduction as well.
2. The diagnostic character of the new species.
The new species has several different characters compared to other Leptopus species, however the leaf size is not clearly distinct as present in the description that the leaf size is 2 – 15 cm x 1.2 – 5 cm. So it does not distinct from the other species that is less than 10 cm. So this character need to be reconsidered.
3. The key to species:
In the couplet 7 and 8, the leaf size was used. It very difficult to separate the species based on leaf size as it was not clearly distinct. So this characters need to be reconsidered.

Annotated reviews are not available for download in order to protect the identity of reviewers who chose to remain anonymous.

---

## Round 0.2 · Minor Revisions

I am sorry for insisting again, but I think that if you explain better in the methods which species you considered for the phylogenetic analyses and the synonyms it will improve the understanding of this section. In addition one of the reviewers suggested some changes.

·

Basic reporting

The authors implement most of the original Review comments and the manuscript is greatly improved editorially. The technical plant description has fleshed out details, but the molecular/phylogenetic methods are still sparse in that regard. A few new editorial comments:

Line numbers here relate to the pdf version.
Line29: “A key to all the Leptopus species in the world is provided.”> ‘A key to all 10 accepted Leptopus species is provided.’ Note: Extraneous to add “in the world”; ‘worldwide’ could be acceptable.
L50: “The species is much different...”> ‘The species is very different...’
L90-91: “To study the phylogenetic position of the new species, a phylogenetic study of the genus Leptopus was performed,”> ‘To resolve the relationships of the new species, a phylogenetic study of the genus Leptopus was performed,’
L154-155: “and mature seeds with copious endosperm (Fig. 4I)”> This detail about copious endosperm also should be added to the seed part (L220) of the plant description, given that it’s stated prior as a generic diagnostic character.
L164-165: “longitudinally strongly ribbed stems), some of its leaves laminas could up to 15 cm long and 7 cm wide (vs. leaf laminas all less than 10 cm long and 3 cm wide), leaves”> ‘longitudinally strongly ribbed stems), laminas up to 15 cm long and 7 cm wide (vs. laminas all less than 10 cm long and 3 cm wide), leaves...”
L172-173: “from the rest four congeneric species”> ’from the remaining four congeneric species’
L217: “2-valed”> ‘2-valved’
L281-282: “Petioles of mature leaves far less than 1/4 of leaf lamina length; leaf apex not rounded (except several specimens of L. chinensis)” > Would it be accurate and cleaner to qualify as ‘Petioles of mature leaves far less than 1/4 of leaf lamina length; leaf apex not rounded (rarely rounded in L. chinensis)’
L242-243: “and L. fangdingianus seems to be reasonable because”> ’and L. fangdingianus is expected because’
L246: “by karst environment”> ‘by a karst environment’
L321: “Leptopus lolonum”> Italics
L327: “Leptopus cordifolius”> Italics

Figure legends need italics for scientific names and plastid genes (presently inconsistent in that regard).

Experimental design

I disagree with the author rebuttal: “The technique of genome skimming is mature enough to date and most of relevant phylogenetic studies did not describe the detailed information about procedures of DNA sequence generation and also some other information, such as the read length, the number of reads, the coverage of the assemblies. So, the method associated with the DNA sequence generation in the present taxonomic study was not described in detailed, but a reference (Zeng et al. 2018) was cited in the manuscript.” I do not accept that no sequencing methods are needed, and most papers do provide the basic information I originally suggested (i.e., how were libraries prepared,
what sequencing platform, read length, how many reads, how were the reads filtered, what was the coverage of the assemblies). While genome skimming may be conceptually well established (or “mature”), how it is executed is very variable today among published studies and improves with time. Skimming with 50 bp reads 5+ years ago is very different from Illumina today or using Pac-Bio, and how Zeng et al. did it 3+ years ago likely differs in details that affect data quality and coverage from what the present authors undertook. Zeng et al. doesn’t even provide basic sequencing details to allow quality to be assessed to begin with, so its citation is not especially informative. While the authors have indicated they plan to do more with the skimming data and may present more details there, the manuscript currently in hand cannot cite that future effort.

L115-117: “All the three datasets were analyzed using two approaches: Bayesian Inference (BI) (Ronquist & Huelsenbeck, 2003) and Maximum Likelihood (ML) (Stamatakis, 2006).” >The references cited are for specific programs and this sentence needs to be rephrased around those programs (i.e., MrBayes v???, RAxML v???) and not merely indicate the BI and ML optimality criteria. Just as in the sequencing methods, attempts here to cite away critical details (in that case to “Yao et al. 2020) fall flat and at least ML BS replicate numbers should be specified here along as noted above, with clarifying the programs used.

Validity of the findings

From the author rebuttal comments: “Actually, we have the GPS coordinates of the two wild populations of the new species, but detailed information about the coordinates are not provided in the manuscript, because of the two populations are not located in nature reserves and some of the new species reported recently in China may be digged after their publication, especially those have good values in medicine or horticulture.” This conservation concern reason regarding not specifying coordinates should be reiterated in the manuscript rather than letting their absence seem like a major lapse. Perhaps along the lines of inserting a parenthetical statement in the type locality, or inserting a statement under ‘Distribution and habitat’ or ‘Ethics statements’. See inserted qualification in this regard in Ethics section: ‘The collection location of the new species reported in this study is outside any natural conservation area and no specific permissions were required for the location. Since this species is currently undescribed, it is not currently included in the China Species Red List (Wang & Xie, 2004). <<However, due to conservation concerns and lack of habitat protection, exact locality coordinates have been withheld from our published specimen records.>> Our field studies did not involve any endangered or protected species. No specific permits were required for the present study.’ (Line 57-61)

---

## Round 0.3 · accepted · Accept

I appreciate that you considered every issue raised in the last round of reviews, I think it improved the reading in the methods.